# The effect of nurses' individual and occupational characteristics and perceptions of hierarchical career plateau on their turnover intention: A cross-sectional study

**Şehrinaz Polat** [ID] [1]*, **Tuğba Yeşilyurt Sevim** [2], **Hanife Tiryaki Şen** [3]

**1** Faculty of Nursing, Nursing Management, Istanbul University, Istanbul, Türkiye, **2** Nursing Department, Nursing Management, Istinye University Faculty of Health Sciences, Istanbul, Türkiye, **3** Hamidiye Faculty of Nursing, Nursing Management, University of Health Sciences, Istanbul, Türkiye

☯ These authors contributed equally to this work.

* polats@istanbul.edu.tr

## Abstract

Career plateau is associated with undesirable work outcomes such as low job performance, motivation, work and organizational commitment, increased counterproductive work behaviors, and turnover intentions. The aim of the present study was to examine the effect of nurses' personal and professional characteristics and their perceptions of hierarchical career plateau on their turnover intentions. Hierarchical plateaus refer to points within an organization's pyramidal structure where opportunities for career advancement become limited or cease entirely. Experiencing a career plateau can result in negative work outcomes, including decreased job satisfaction and reduced motivation. This cross-sectional study was conducted from March to June 2024 with 464 nurses working in hospitals across Türkiye. Participants were selected using a convenience sampling method. Data were collected using a Nurse Information Form, the Career Plateau Survey and Turnover Intention Scale. Multivariate linear regression analysis was used to determine the effect of independent variables (individual and occupational characteristics and perceptions of hierarchical career plateau) on the dependent variable (turnover intention). This study was conducted in accordance with the STROBE checklist for cross-sectional studies. The findings indicated that nurses' turnover intention increased as the level of hierarchical career plateau increased. Additionally, nurses working in private/non-profit healthcare organizations who had voluntarily chosen their profession exhibited statistically significantly lower turnover intentions. Interventions aimed at reducing career plateaus, tailored to the type of healthcare institution, and supporting nurses' autonomy in career choice can reduce turnover intentions. It is recommended to regularly review promotion policies in healthcare institutions to align with the needs and expectations of current nurses, and to implements programs specific to their career development.

**Data availability statement:** All relevant data are within the paper and its Supporting Information files.

**Funding:** The author(s) received no specific funding for this work.

**Competing interests:** The authors have declared that no competing interests exist.

## Introduction

Turnover intention is defined as an employee's reported willingness to leave the organization within a defined period of time [1], which includes a conscious, deliberate [2] and multi-stage process [3]. It is a strong predictor of actually leaving the institution [3–8]; and therefore, turnover intention is discussed in the literature more frequently rather than actual nurse turnover behavior. An increase in an employee's turnover intention indicates that this employee is nearing the decision to leave the organization [4]. Identifying the intention, antecedents, and consequences of nurse turnover is essential for developing effective retention strategies [9]. A recent systematic review and meta-analysis has reported that the global cumulative prevalence of nurse turnover is 18% [10]. Another meta-analysis reports that the global nurse turnover rate ranges from 8% to 36.6%, with a combined rate of 16% [11]. The nursing shortage and its associated negative consequences remain its importance as a global issue [11]. Numerous studies have demonstrated that nurses' turnover intention and the resulting nursing shortage have various adverse effects on patients, nurses, and healthcare organizations [12,13].

A systematic review classifies the causes of nurse turnover into four broad categories: individual factors (e.g., sociodemographic characteristics such as age, education, and sex, as well as psychological experiences as stress and burnout), work-related factors (e.g., workload, role conflict, autonomy, shift patterns, and career or promotion expectations), interpersonal factors (e.g., relationships with leaders and colleagues), and organizational factors (e.g., work environment and salary) [6]. Among the work-related factors, career plateau has been identified as a key contributor to nurses' high turnover intention and actual turnover behavior [14]. Research indicates that nurses and healthcare professionals often perceive career plateau as widespread within the profession [15–20]. Career plateau is commonly recognized as a subjective measure of employees' perceptions of their career progression [21]. Therefore, reducing nurses' perceptions of career plateau may play a critical role in reducing turnover intentions.

The concept of career plateau was first introduced by Ference et al. [22], who defined it as "the point at which the probability of additional hierarchical promotion in the career is very low" (p.602). Initially, career plateau was considered as a one-dimensional concept, primarily based on objective criteria such as age and duration of experience in the organization [22]. However, subsequent research has shown that career plateau is more accurately described as a two-dimensional construct, which includes both subjective perceptions—such as job content plateau—and objective factors, such as hierarchical career plateau [22,23]. This two-dimensional framework suggests a more nuanced understanding of career development and management. Hierarchical plateaus refer to points within the organizational structure, particularly in hierarchical or pyramidal systems, where opportunities for promotion significantly diminish or are altogether absent. In this case, after employees reach a certain level, their opportunities for promotion are restricted or become unachievable. Job content plateaus refer to points at which employees have fully mastered their jobs and experience the perception that they no longer have any challenges in their work. In this case, employees who have reached the job content plateau perform their work in a routine manner and no longer have the opportunity to develop or progress [23].

According to Farivar et al. [24], individuals may not always view their career plateau as a negative experience, and some may intentionally choose to remain in this state. This decision may be associated with factors such as work-life conflict, increasing job demands, the potential for heavier workloads, job satisfaction, or the belief that further promotion opportunities are unlikely for various reasons [24]. Since career plateau is a subjective perception, an individual's positive or negative perception of their situation can significantly influence their future experiences, attitudes, and behaviors. The experience of career plateau is likely to have

a more pronounced negative impact on work outcomes for nurses who attempt to advance yet encounter limited promotion opportunities.

The literature suggests that hierarchical plateau is generally considered a stressful experience [25] and is regarded as a workplace stressor that triggers negative emotions [26]. In a study by Ng and Yang [26], employees who experience career plateau are more likely to encounter negative emotions, particularly when they perceive the plateau as a threatening stressor—such as when they are eager to progress in their careers yet feel blocked or constrained in achieving advancement. These individuals are also more likely to engage in negative behaviors as a coping mechanism to alleviate their perceived emotional discomfort.

Some potential precursors of career plateau include positive attitudes toward one's job, career alignment, perceptions of organizational support, collegial support, supervisor support, and sociodemographic factors such as age, marital status, sex, educational level, and years of experience [19,21,23]. Factors influencing nurses' perceptions of career refer to individual characteristics, limited career advancement opportunities, organizational influences, and personal experiences [14]. Research has shown that individuals experiencing career plateau often report negative emotional outcomes, which are associated with undesirable job-related consequences, including decreased job performance, job satisfaction, motivation and commitment to both work and the organization, poorer extra-role performance, increased counterproductive work behaviors, and higher turnover intention [21,25–27]. Huyghebaert et al. [28] have found that nurses are less inclined to leave their organizations when they perceive available career opportunities. In studies conducted on nurses, career plateau is consistently associated with increased turnover intention [14,29]. However, despite some research associating career plateau with nurses' turnover intention, there has been insufficient attention to this issue, with a notable lack of relevant studies at both regional and national levels in Türkiye.

It is essential to explore the impact of career plateau on nurses' turnover intention and to expand the current literature by incorporating additional research from diverse cultural and geographical contexts. This study aimed to examine the effect of nurses' individual and occupational characteristics, as well as their perceptions of hierarchical career plateau, on their turnover intention.

## Materials and methods

### Design

This study is a cross-sectional descriptive design and adheres to the STROBE (Strengthening the Reporting of Observational Studies in Epidemiology) checklist for cross-sectional studies.

### Sample and setting

The study was conducted across Türkiye, with the target population consisting of nurses working in hospitals who were directly responsible for patient care. The minimum sample size was calculated using a sampling method based on a known population. The formula $n = \frac{Nt^2pq}{d^2(N-1) + t^2pq}$ was applied, with a 95% confidence interval, a 5% margin of error, and a 50% incidence (p) rate, which represents the maximum value for sample size calculations. The sample size calculation was based on the total number of nurses actively working in Türkiye (243,565), as reported in the 2022 statistical yearbook of the Ministry of Health. This calculation determined that a minimum sample size of 384 nurses was required to achieve a 95% confidence level and a 5% margin of error. A "required to answer" sign was added to all questions in the questionnaire. Nurses participating in the study could not proceed to the next question or submit the form without responding the previous one. A total of 464 nurses who completed the questionnaire constituted the sample of the study.

The participants were selected using convenience sampling. The inclusion criteria were: (1) working directly in patient care in a hospital in Türkiye, (2) having been employed at the same institution for at least one year, and (3) voluntarily agreeing to participate in the research. The first page of the questionnaire included three questions to verify whether nurses met these criteria: working directly in patient care at a hospital, had been employed at the same institution for at least one year, and voluntarily agreed to participate in the study. Participants were not allowed to proceed without responding these questions. A random sampling method was used to select the nurses included in the final sample.

## Measurement

**Nurse information form.** The form consisted of a total of 12 questions regarding the nurses' demographic and occupational characteristics, including: age, sex (female/male), marital status (married/single), educational level (nursing high school diploma/bachelor's degree in nursing/postgraduate), unit (ward/operating room/intensive care/other), work schedule (fixed days/constant nights/shifts), type of institution (public/private-foundation), position (nurse/management), whether they willingly chose the profession (yes/no), duration of professional experience, total duration of experience in their current unit, and total duration of experience in their current position.

**Career plateau survey.** The Career Plateau Survey was developed by Milliman [30], and its Turkish validity and reliability study was conducted by Ak and Soybali [31]. It is a self-report scale consisting of 12 questions and two subscales. The first six questions assess job content career plateau, while the last six measure hierarchical career plateau. The scale is rated on a 6-point Likert-type scale (1: Strongly Disagree, 2: Disagree, 3: Partially Disagree, 4: Partially Agree, 5: Agree, 6: Strongly Agree). Responses to the first, second, fourth, fifth, sixth, eighth, and twelfth statements are reverse scored. For this study, the hierarchical career plateau subscale was used. Higher mean scores indicate a higher level of perceived hierarchical career plateau, while lower mean scores indicate a lower level. The minimum possible score on the scale is 6, and the maximum possible score is 36. In the Turkish adaptation of the scale, Cronbach's alpha reliability coefficient for the hierarchical career plateau subscale is 0.86 [31], and it was found to be 0.928 in this study.

**Nurse turnover intention scale.** The scale, developed by Yeun and Kim [32] and adapted to Turkish by Zeyrek et al. [33], is a 5-point Likert-type scale (Strongly Disagree = 1, Disagree = 2, Neutral = 3, Agree = 4, Strongly Agree = 5). It is unidimensional and does not include any reverse-coded items. The total score on the scale ranges from 10 to 50, with higher scores indicating a higher level of turnover intention. Cronbach's alpha coefficient for the scale is 0.83 [33], and it was found to be 0.916 in this study.

## Data collection

Data were collected between March 4th and May 14th, 2024. The questionnaire was created using Google Forms, and the link was shared through social media platforms such as Instagram, Facebook, and WhatsApp. Nurses who responded to the questionnaire were included in the sample.

## Ethical considerations

This study was conducted in accordance with the Declaration of Helsinki, and all procedures involving human participants were approved by the Istanbul University Social Sciences and Humanities Research Ethics Committee (Approval Date: February 12, 2024, Approval Number: 2415631). Permission to use the Turkish version of the adapted scales was obtained

via email from the original authors. A consent form was provided on the first page of the questionnaire, where participants were asked to choose between "I agree to participate in the study" or "I do not agree to participate in the study." Only participants who chose "I agree to participate in the study" were allowed to proceed with the questionnaire.

## Statistical analysis

In this study, the normality of continuous variables was assessed using the Kolmogorov-Smirnov test. Categorical variables were presented as frequency (n, %) and continuous variables as mean and standard deviation. Cronbach's alpha coefficients were calculated to assess the reliability of the scales. Comparisons between two groups for continuous variables were performed using an independent samples t-test, while one-way analysis of variance (ANOVA) was used for comparisons involving more than two groups. Pearson's correlation test was used to examine the correlation between two continuous variables. Multivariate linear regression analysis was conducted to assess the effect of independent variables on the dependent variable (turnover intention). The results were evaluated with 95% confidence intervals, and statistical significance was set at $p < 0.05$. No data were missing for the statistical analyses. The sample size was determined using the known population sampling formula. All statistical analyses were performed using SPSS software, version 26 (IBM Corp., Armonk, NY, USA).

## Results

### Characteristics of the participants

A total of 464 nurses participated in the study, with an average age of $34.74 \pm 8.31$ years, 82.5% of whom were female. Regarding demographic characteristics, 71.3% of the nurses were married, and 60.6% held an undergraduate degree. In terms of occupational characteristics, 37.1% worked in a ward, 32.5% in the operating room, 17.7% in the intensive care unit, and 57.5% in private or foundation hospitals. Additionally, 79.3% of participants were working as nurses, with 40.1% on constant day shifts, 31.9% on constant night shifts, and 28% working rotating shifts. The mean total professional experience, years of experience in their current unit, and years of experience in their current position were $12.93 \pm 8.03$, $10.36 \pm 5.95$, and $11.89 \pm 7.59$ years, respectively (Table 1).

### Descriptive information of hierarchical career plateau and turnover intention

The mean level of hierarchical career plateau among nurses was $23.03 \pm 7.29$ (range: 8–35), while the mean turnover intention level was $34.61 \pm 9.37$ (range: 13–47). A statistically significant positive correlation was found between nurses' hierarchical career plateau level and their turnover intention, with turnover intention increasing as the level of hierarchical career plateau increased ($r = 0.821$; $p < 0.001$) (Table 2).

### Univariate analysis results

No correlation was found between nurses' age, length of experience in the position, length of experience in the unit, and length of professional experience and turnover intentions. Nurses who worked in private/foundation health institutions ($t = 5.013$; $p < 0.001$) and enjoyed their profession ($t = 10.824$; $p < 0.001$) were found to have statistically significantly lower turnover intention. No statistically significant difference was found in turnover intention according to sex, marital status, educational level, unit, work schedule and position of nurses ($p > 0.05$) (Table 3).

**Table 1. Profile of the respondents.**

| Variables | Category | Scores (Mean±SD) | n | % |
|---|---|---|---|---|
| **Age** | N/A | 34.74 ± 8.31 | N/A | N/A |
| **Duration of professional experience (years)** | N/A | 12.93 ± 8.03 | N/A | N/A |
| **Total duration of experience in the unit (year)** | N/A | 10.36 ± 5.95 | N/A | N/A |
| **Total duration of experience in the position (year)** | N/A | 11.89 ± 7.59 | N/A | N/A |
| **Gender** | Female | N/A | 383 | 82.5 |
| | Male | N/A | 81 | 17.5 |
| **Marital status** | Married | N/A | 331 | 71.3 |
| | Single | N/A | 133 | 28.7 |
| **Educational level** | High school | N/A | 37 | 8.0 |
| | Associate degree | N/A | 77 | 16.6 |
| | Bachelor's degree | N/A | 281 | 60.6 |
| | Postgraduate | N/A | 69 | 14.9 |
| **Unit** | Ward | N/A | 172 | 37.1 |
| | Operating room | N/A | 151 | 32.5 |
| | Intensive Care | N/A | 82 | 17.7 |
| | Other | N/A | 59 | 12.7 |
| **Work schedule** | Constant days | N/A | 186 | 40.1 |
| | Constant nights | N/A | 148 | 31.9 |
| | Shifts | N/A | 130 | 28.0 |
| **Type of institution** | Public | N/A | 197 | 42.5 |
| | Private/Foundation | N/A | 267 | 57.5 |
| **Position** | Nurse | N/A | 368 | 79.3 |
| | Nurse Manager | N/A | 96 | 20.7 |
| **Willingly choosing the profession** | Yes | N/A | 214 | 46.1 |
| | No | N/A | 250 | 53.9 |

SD: Standard Deviation, N/A: Not Available.

**Table 2. Hierarchical career plateau and turnover intention scale scores and the level of correlation between scales.**

| Measurements | Scores (Mean±SD) | Minimum | Maximum | r | *P*-value |
|---|---|---|---|---|---|
| **Hierarchical Career Plateau** | 23.03 ± 7.29 | 8 | 35 | 0.821 | <0.001* |
| **Nurse Turnover Intention Scale** | 34.61 ± 9.37 | 13 | 47 | | |

*p < 0.01, r: Pearson's correlation test, SD: Standard deviation.

## Multivariate analysis results

Multivariate linear regression analysis was conducted to identify the independent variables associated with nurses' turnover intention (Table 4). The model explained approximately 70.5% of the variance in the total turnover intention score. An increased hierarchical career plateau (β = 0.738; p < 0.001) and working in a public institution (β = 0.189; p < 0.001) were found to have a positive effect on turnover intention, while willingly choosing the nursing profession (β = -0.349; p < 0.001) had a preventive effect on turnover intention (Table 4).

**Table 3. The level of nurses' turnover intention according to the introductory. Characteristics.**

| Variables | Category | n | Turnover intention | | |
|---|---|---|---|---|---|
| | | | Scores (Mean±SD) | Test value | P-value |
| **Age** | All | 464 | N/A | 0.016[c] | 0.727 |
| **Duration of professional experience** | All | 464 | N/A | 0.030[c] | 0.526 |
| **Duration of experience in the unit** | All | 464 | N/A | 0.023[c] | 0.624 |
| **Duration of experience in the position** | All | 464 | N/A | 0.033[c] | 0.484 |
| **Gender** | Female | 383 | 34.69 ± 9.24 | 0.388[a] | 0.698 |
| | Male | 81 | 34.25 ± 10.03 | | |
| **Marital status** | Married | 331 | 34.71 ± 9.34 | 0.347[a] | 0.729 |
| | Single | 133 | 34.38 ± 9.48 | | |
| **Educational level** | High school | 37 | 35.86 ± 8.49 | 1.143[b] | 0.331 |
| | Associate degree | 77 | 35.70 ± 8.92 | | |
| | Bachelor's degree | 281 | 34.51 ± 9.37 | | |
| | Postgraduate | 69 | 33.14 ± 10.24 | | |
| **Unit** | Ward | 172 | 34.77 ± 9.57 | 0.168[b] | 0.918 |
| | Operating room | 151 | 34.81 ± 9.23 | | |
| | Intensive Care | 82 | 33.96 ± 9.07 | | |
| | Other | 59 | 34.58 ± 9.73 | | |
| **Working system** | Constant days | 186 | 35.19 ± 8.82 | 2.629[b] | 0.073 |
| | Constant nights | 148 | 33.17 ± 10.14 | | |
| | Shifts | 130 | 35.43 ± 9.09 | | |
| **Type of institution** | Public | 197 | 36.99 ± 7.67 | **5.013[a]** | **<0.001*** |
| | Private/Foundation | 267 | 32.86 ± 10.11 | | |
| **Position** | Nurse | 368 | 34.93 ± 9.15 | 1.350[a] | 0.179 |
| | Nurse Manager | 96 | 33.40 ± 10.13 | | |
| **Willingly choosing the profession** | Yes | 214 | 29.93 ± 10.28 | **10.824[a]** | **<0.001*** |
| | No | 250 | 38.62 ± 6.14 | | |

*p < 0.01, SD: Standard Deviation, N/A: Not Available,

[a]: Independent sample t-test,

[b]: One-way ANOVA,

[c]: Pearson's correlation test.

**Table 4. Factors influencing nurses' turnover intention.**

| Variables | B | SE | β | t | P-value | R² | Adjusted R² | F |
|---|---|---|---|---|---|---|---|---|
| **Constant term** | 13.556 | 0.960 | | 14.110 | <0.001* | 0.707 | 0.705 | F = 369.717 |
| **Hierarchical career plateau** | 0.947 | 0.035 | 0.738 | 26.459 | <0.001* | | | p < 0.001* |
| **Working at a public institution** | 1.772 | 0.485 | 0.189 | 3.648 | <0.001* | | | |
| **Willingly choosing the profession** | -3.272 | 0.515 | -0.349 | -6.342 | <0.001* | | | |

*p < 0.01, B: Estimates of unstandardized standardized regression weights, β: Estimates of standardized regression. weights. SE: Standard error, R²: Coefficient of determination.

## Discussion

In the present study, the effect of nurses' individual and occupational characteristics and their perceptions of hierarchical career plateau on turnover intention was examined. The findings revealed that nurses' perceptions of a higher hierarchical career plateau were associated with

increased turnover intention, while working in a private hospital and having a high level of job satisfaction were found to decrease turnover intention.

## Discussion of hierarchical plateau and turnover intention averages

The average hierarchical career plateau level of nurses in this study was found to be high, consistent with findings from the literature [15,18,20]. However, there are also a limited number of studies that report contrary results [15]. This difference can be explained by the fact that career plateau is both an objective and subjective phenomenon, influenced by personal characteristics, organizational factors, and the opportunities and privileges available to individuals, as well as their perceptions of these opportunities [17]. In a study by Zhu and Li [20] on career plateau among 2,680 senior nurses, it is reported that the highest incidence rate is associated with hierarchical plateau, which one of the dimensions of the career plateau. As reported in the literature, this finding may be attributed to the hierarchical structure of organizations, which often resemble a pyramid. Consequently, the number of promoting nurses is limited, reflecting the reduced likelihood of vertical progression within the organization. Therefore, optimizing job positions and offering more promotion opportunities and career development paths may help alleviate the hierarchical career plateau experienced by nurses.

In this study, nurses' turnover intention was found to be above average. Similar studies conducted in Türkiye have also reported moderate to high levels of turnover intention among nurses [34,35]. Some studies in the literature have shown that nurse turnover rates vary across continents and that this is related to differences in health systems, working conditions, and health policies between developed and developing countries [10,21,36].

## Discussion of the impact of willingly choice of careers on turnover intentions

It was found that nurses who willingly chose the nursing profession had lower turnover intention. Nurses may choose the nursing profession for various reasons, including personal interest, perception of the professional image, a desire to help others, job security, favorable living conditions, recommendations from family, friends, and teachers, opportunities for migration abroad, family pressure, parental authority, social influences, or even chance (such as placement based on university exam scores) [37–42]. A study conducted in China found that nurses who were assigned to their job roles or chose their careers based on the wishes of their parents did not demonstrate expertise in their current profession and were less interested in their work. Additionally, this study reported that 72.5% of retired nurses who left the profession had not chosen nursing willingly [43]. Similarly, a study by Yeşilyurt et al. [44] conducted across Türkiye found that nurses who did not willingly choose the profession had higher turnover intentions [44]. Another study conducted with nurses in a public hospital in Türkiye also reported that those who did not willingly choose the nursing profession had higher turnover intentions compared to those who chose willingly [34]. A study conducted among nursing students in Türkiye found that two-thirds of the students chose nursing primarily due to job security [42]. Nurses who willingly enter the profession and have a passion for their work tend to focus on the positive aspects of their role, rather than the negative factors such as challenging working conditions. This mindset helps them to remain resilient against the prevailing organizational challenges, fostering a long-term commitment to their profession [45]. Adib-Hajbaghery et al. [45] emphasize that a lack of passion for the nursing profession can increase the likelihood of nurses distancing themselves from their professional roles, experiencing burnout, and ultimately leaving the profession [45]. Nurse managers should develop strategies to enhance nurses' interest in the profession and offer career counseling.

Additionally, before entering nursing education, prospective nursing students should receive proper information about the profession, and support should be provided to help them explore their areas of interest within nursing.

## Discussion of the relationship between working in a private or public hospital and turnover intentions

According to the results of the study, it was found that working in public institutions increased nurses' turnover intention. While similar findings have been reported in studies conducted in Türkiye [46], other studies have indicated that nurses working in private hospitals have higher turnover intentions compared to those in public hospitals [2,44,46,47].

The literature indicates that the type of employment (public or private) plays a significant role in determining job security, with nurses in government positions reporting significantly higher job security [48]. Studies showing lower turnover intentions among nurses in public hospitals attribute this finding to the greater job security offered by public organizations [2,48]. Huaman Ramirez and Lahlouh [23] also suggest that employees who choose to work in public organizations may be less concerned with advancing in hierarchical positions and may prioritize other factors, such as job security, more highly [23]. In the present study, it was observed that nurses in public hospitals tend to work longer due to job security, which may contribute to the lower frequency of managerial turnover. This stability in management could, in turn, influence the higher turnover intention among nurses. Future studies may explore variables such as job security through incorporating into the model for further research.

It has been reported in the literature that a more positive perception of the work environment reduces turnover intentions [49]. Studies that report high turnover intentions in the public sector often attribute these intentions to negative working conditions, such as ineffective communication, inadequate compensation, poor management, availability of resources, participation in decision-making, transformational leadership, low organizational support, collegial and superior support, procedural justice, lack of career advancement opportunities, and inadequate staffing [50–52]. In private hospitals, it has been reported that turnover intention decreases in institutions that have more positive work environment factors such as perceived managerial support, and less negative factors such as occupational stress and work-family conflict [50]. In this context, there has been a growing demand for better working conditions, particularly in the public sector, accompanied by a rising turnover intention [2]. Additionally, the mandatory nurse-to-patient ratio standards in private hospitals, particularly for inpatient services, contrasts with the lack of enforceable standards for nurse-patient ratios in public hospitals, except in specialized units such as intensive care. This discrepancy may have led to increased workload in public hospitals, contributing to higher turnover intention. Furthermore, while the recruitment of new nurses in public hospitals may require several procedures and government approval to open new positions, the faster hiring process in private hospitals may have resulted in a heavier workload for nurses in private institutions, which increased turnover intention in these settings.

One study highlights that additional support systems, such as transportation, childcare, housing, and private health insurance, offered in private hospitals, are considered critical for employee retention. In contrast, the absence of such supports in public hospitals is identified as a significant reason for nurses' turnover behavior [47]. In the present study, the lack of additional support services, such as childcare, housing, and transportation services for each shift, which are commonly offered in private hospitals, may have contributed to higher turnover intention in public hospitals.

In a study comparing management practices in public and private hospitals, it is reported that public hospitals appear to be less efficient in promoting and rewarding highly motivated employees or providing incentive plans to remove low performers [53]. In a meta-analysis by Hur and Abner [51] of studies conducted on the reasons for turnover in public organizations, promotion, diversity management, performance appraisal/feedback, satisfaction with practices such as training/development and salary and rewards have a negative association with turnover intention [51] A qualitative study investigating the reasons for turnover and retention of nurses in private and public hospitals in Türkiye emphasizes that the lack of objective criteria for promotion and inequalities in promotion in public hospitals are associated with turnover, while faster hierarchical advancement and merit-based advancement are important for retention in private hospitals compared to public hospitals [47]. Freire and Azevedo [2] have found that normative commitment has a direct and negative significant effect on turnover intention of nurses in public hospitals, while affective commitment has a direct and negative significant effect on turnover intention of nurses in private hospitals [2]. In public hospitals, nurses' turnover intention may be high due to several factors, including normative commitment, long-term employment of nurse managers driven by job security, and slow hierarchical progression. Other contributing factors include the perceived lack of fairness in promotion decisions, unclear promotion criteria, and the prioritization of seniority over merit when making promotion decisions. These elements may lead to frustration and decreased job satisfaction, ultimately increasing turnover intention among nurses in public hospitals.

The findings of this study suggest that the quality of the private hospitals where the nurses in the sample work may also play a role in influencing their turnover intentions. Some private hospitals in Türkiye are accredited and offer favorable working conditions, good physical infrastructure, and additional benefits, which can contribute to higher job satisfaction and lower turnover intentions. These results highlight the significant impact that the type of hospital—whether public or private—has on the facilities and benefits provided to employees, ultimately influencing their turnover intention. For healthcare organizations and managers, it is necessary to diagnose the situation and define the most appropriate policies for each sector as a strategy to retain healthcare workers.

## Discussion of the relationship between career plateau and turnover intention

In the present study, it was found that as the level of hierarchical career plateau of nurses increased, their level of turnover intention also increased. This finding is consistent with previous research in various sectors [27,54] as well as in nursing [14,29], which has shown a positive correlation between hierarchical plateau and turnover intention. In other words, a higher perceived hierarchical plateau tends to lead to an increased turnover intention. A study by Ng et al. (2024) has found that employees with high career plateaus had a weaker negative correlation in career planning and turnover intention, while those with high level of career plateau have higher turnover intentions [55]. A systematic review and meta-analysis has also reported that turnover intention among nurses is significantly associated with promotion and career development opportunities [56]. Turnover intention often results from the negative experiences encountered by employees. In a qualitative study by Zhu et al. [14], which examined the career plateau experiences of nurses who resigned due to limited career advancement opportunities, all participants stated that they did not make a direct decision to resign due to their perception of being stuck. However, they had previously faced restricted opportunities to advance their careers, leading to negative emotions such as uncertainty, unhappiness, disappointment, and job dissatisfaction. As a result, these negative feelings caused nurses to lose confidence in their current organization and seek employment elsewhere, with better

prospects for career growth and promotion [14]. Similarly, Yeşilyurt et al. [44] found that nurses dissatisfied with the career opportunities provided by their hospital exhibited a higher level of turnover intention [44]. De Clercq et al. [57] also noted that when employees perceive their employers as providing insufficient career support, they may feel that their employers are not meeting their obligations, leading to feelings of frustration and dissatisfaction.

According to the meta-analytical review by Lu et al. [58], the deterioration of the psychological contract between nurses and their employers negatively impacts turnover intention. The Social Exchange Theory suggests that when employees perceive their employer as failing to fulfill their obligations, they may feel justified in not fulfilling their own, which can lead to job dissatisfaction and ultimately, turnover. Based on the findings of the present study, nurses may intend to leave their positions to seek better promotion opportunities and other resources that they feel are unavailable in their current workplace.

## Limitations

This study employed a cross-sectional design. Future research may explore the relationship between career plateau and turnover intention using longitudinal studies to better understand the causal dynamics over time. Additionally, the scope of independent variables in the current study may be limited; therefore, future studies may incorporate a broader range of variables to provide a more comprehensive analysis. Another limitation of this study is that the data were based on self-reports from nurses, which may introduce bias or inaccuracies in responses.

## Conclusions

This study found that factors such as career plateau, type of hospital, and the voluntary choice of the nursing profession significantly influenced nurses' turnover intention. The findings provide valuable insights for hospitals and nurse managers to enhance human resource management, particularly in addressing the key determinants of turnover intention and career plateau in nursing. When promotable positions are limited in hospitals, offering alternative career development opportunities can help reduce turnover intentions, which are required to be more appealing compared to traditional promotions. Furthermore, providing career counseling to nursing candidates during their entry into nursing education can help foster greater interest and motivation in the profession, ultimately reducing turnover intentions. Students who are more passionate about their chosen career are likely to have lower turnover intentions once they begin their professional practice. The influence of hospital type on turnover intentions underscores the importance of developing strategies to improve the working environment, opportunities, and employee rights. Healthcare organizations can enhance nurse retention, improve job satisfaction, and reduce turnover rates through focusing on these areas.

## Supporting information

**S1 Data. Data set_v2.**
(7Z)

## Acknowledgments

We wish to thank the nurses who participated in this study.

## Author contributions

**Conceptualization:** Şehrinaz Polat.

**Data curation:** Şehrinaz Polat, Tuğba Yeşilyurt Sevim.

**Formal analysis:** Şehrinaz Polat.

**Funding acquisition:** Tuğba Yeşilyurt Sevim, Hanife Tiryaki Şen.

**Methodology:** Şehrinaz Polat, Hanife Tiryaki Şen.

**Resources:** Şehrinaz Polat, Tuğba Yeşilyurt Sevim, Hanife Tiryaki Şen.

**Supervision:** Şehrinaz Polat.

**Validation:** Şehrinaz Polat.

**Visualization:** Şehrinaz Polat.

**Writing – original draft:** Şehrinaz Polat, Tuğba Yeşilyurt Sevim, Hanife Tiryaki Şen.

**Writing – review & editing:** Şehrinaz Polat.

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
