## [Decision Letter · Decision Letter 0]

18 Oct 2024

PONE-D-24-39377The effect of nurses' individual and occupational characteristics and perceptions of hierarchical career plateau on their turnover intention: A cross-sectional study

PLOS ONE

Dear Dr. POLAT,

Thank you for submitting your manuscript to PLOS ONE. After careful consideration, we feel that it has merit but does not fully meet PLOS ONE’s publication criteria as it currently stands. Therefore, we invite you to submit a revised version of the manuscript that addresses the points raised during the review process.

Kindly provide a summary of the conclusion. It is excessively lengthy. 

We look forward to receiving your revised manuscript.

Kind regards,

Roaa Sabri Gassas

Academic Editor

PLOS ONE

2. In this instance it seems there may be acceptable restrictions in place that prevent the public sharing of your minimal data. However, in line with our goal of ensuring long-term data availability to all interested researchers, PLOS’ Data Policy states that authors cannot be the sole named individuals responsible for ensuring data access (http://journals.plos.org/plosone/s/data-availability#loc-acceptable-data-sharing-methods).

Reviewers' comments:

Reviewer's Responses to Questions

**Comments to the Author**

1. Is the manuscript technically sound, and do the data support the conclusions?

Reviewer #1: Yes

Reviewer #2: Yes

Reviewer #3: Yes

Reviewer #4: Yes

2. Has the statistical analysis been performed appropriately and rigorously? 

Reviewer #1: Yes

Reviewer #2: Yes

Reviewer #3: Yes

Reviewer #4: Yes

3. Have the authors made all data underlying the findings in their manuscript fully available?

Reviewer #1: Yes

Reviewer #2: Yes

Reviewer #3: Yes

Reviewer #4: Yes

4. Is the manuscript presented in an intelligible fashion and written in standard English?

Reviewer #1: No

Reviewer #2: Yes

Reviewer #3: Yes

Reviewer #4: Yes

5. Review Comments to the Author

Reviewer #1: 1. The first line of abstract should be start with the background of the study. It is directly started from the objective which seems inappropriate.

2. The line starting from Data were collected…. Includes both the methodology and analysis which should be separate as in the next line it is written that multivariate linear regression analysis was used….. so the author can remove the “anlalyzed using linear regression” term in the previous line.

3. Independent variables should be mentioned in the abstract as the dependent variable is mentioned here.

4. In introduction part nothing is mentioned regarding the turnover intention. Firstly, the author should mention about what is turnover intention and then try to link it with the literature reviews.

5. There is no information about the non-respondents or non-response item. The author should mention about the sample size after excluding the non-responses also in the study. Is all 464 sampled nurses responded to the form?

6. Table 1 title should be Profile of the respondents

7. In table 2, *p<0.001 description is not mentioned. It should be p<0.01 at 1% level of significance.

8. The variables of table 3 should be described in a brief and proper way.

9. Multivariate analysis is not clear as it is totally messed up and not indicates the concise clear result. Rewrite it.

10. Reformat the table 4. The result of linear regression model formatting is totally weird. Alignment and proper capitalization of words are necessary.

11. In the discussion part, the line starting with “In a study ……….” Is wrong. This study is conducted by the author that’s why it should be stated like “ In the present study, it was tried to examine………” and also write all the results and discussion part in past tense as this study was already completed.

12. Many times “ In the study” phrase is used everywhere, try to rewrite the sentences with proper alignment and sentence reframing.

13. Write all the results and introduction or all over content in your own words or refer to standard paper language. This paper uses too much use AI tool such as Chatgpt.

Reviewer #2: Overall, this manuscript is good, but several things need to be considered, such as the period of the reference sources used. I think it is better to use references within the last 5 years. The manuscript is great and the data support the conclusions. The statistical analysis has been performed appropriately and rigorously. The authors have made all data underlying the findings in their manuscript fully available. The manuscript is presented in an intelligible fashion and written in standard English.

Reviewer #3: The study provides valuable insights into the impact of career plateaus on nurses’ turnover intention.

Given the above, consider using thematic subheadings in the discussions to reflect the study’s results and enhance the readability of the discussion.

Again, consider expanding the discussion on how different hospital types influence turnover intention.

Please ensure that all references are up-to-date and relevant to the study’s context, while you also review for some minor grammatical corrections entirely.

Concerning formatting of the work, please ensure that the tables and work are well formatted to reduce the occurrence of blank spaces and tables spilling over to other pages.

Thank you.

Reviewer #4: The research is conducted in a scientific and organized manner, contains the research methodological conditions, and is accepted for publication.I commend the effort made in the research and the organized work and achievement by the researchers of the research.

6. PLOS authors have the option to publish the peer review history of their article (what does this mean? ). If published, this will include your full peer review and any attached files.

**Do you want your identity to be public for this peer review?** For information about this choice, including consent withdrawal, please see our Privacy Policy .

Reviewer #1: No

Reviewer #2: No

Reviewer #3: No

Reviewer #4: **Yes: ** Mohammed Musaed AL-Jabri

---

## [Author Response · Author response to Decision Letter 0]

20 Nov 2024

I would like to thank the reviewers for their suggestions and criticisms. We have made the necessary arrangements by accepting the suggestions of all the reviewers.

Sincerely yours

---

## [Decision Letter · Decision Letter 1]

18 Dec 2024

The effect of nurses' individual and occupational characteristics and perceptions of hierarchical career plateau on their turnover intention: A cross-sectional study

PONE-D-24-39377R1

Dear Dr. ŞEHRİNAZ POLAT

We’re pleased to inform you that your manuscript has been judged scientifically suitable for publication and will be formally accepted for publication once it meets all outstanding technical requirements.

Kind regards,

Roaa Sabri Gassas

Academic Editor

PLOS ONE

Additional Editor Comments (optional):

Reviewers' comments:

Reviewer's Responses to Questions

**Comments to the Author**

1. If the authors have adequately addressed your comments raised in a previous round of review and you feel that this manuscript is now acceptable for publication, you may indicate that here to bypass the “Comments to the Author” section, enter your conflict of interest statement in the “Confidential to Editor” section, and submit your "Accept" recommendation.

Reviewer #1: All comments have been addressed

Reviewer #3: All comments have been addressed

2. Is the manuscript technically sound, and do the data support the conclusions?

Reviewer #1: Yes

Reviewer #3: Yes

3. Has the statistical analysis been performed appropriately and rigorously? 

Reviewer #1: Yes

Reviewer #3: Yes

4. Have the authors made all data underlying the findings in their manuscript fully available?

Reviewer #1: Yes

Reviewer #3: Yes

5. Is the manuscript presented in an intelligible fashion and written in standard English?

Reviewer #1: Yes

Reviewer #3: Yes

6. Review Comments to the Author

Reviewer #1: The previous comments have been addressed, and now this research paper is acceptable on the grounds of ethics and useful insight from research study.

Reviewer #3: The authors have addressed all the comments by the reviewers. the manuscript is well presented and sound.

7. PLOS authors have the option to publish the peer review history of their article (what does this mean? ). If published, this will include your full peer review and any attached files.

**Do you want your identity to be public for this peer review?** For information about this choice, including consent withdrawal, please see our Privacy Policy .

Reviewer #1: No

Reviewer #3: No

---

## [Editor Report · Acceptance letter]

PONE-D-24-39377R1

PLOS ONE

Dear Dr. Polat,

I'm pleased to inform you that your manuscript has been deemed suitable for publication in PLOS ONE. Congratulations! Your manuscript is now being handed over to our production team.

Kind regards,

on behalf of

Dr. Roaa Sabri Gassas

Academic Editor

PLOS ONE